# Minimally Invasive Conversion Surgery for Unresectable Gastric Cancer with Splenic Metastasis and Splenic Vein Tumor Thrombus: A Case Report

Nobuhisa Tanioka [1,*] , Michio Kuwahara [1], Takashi Sakai [1], Yuzuko Nokubo [1], Shigeto Shimizu [1], Makoto Hiroi [2] and Toyokazu Akimori [1]

1   Department of Surgery, Hata Kenmin Hospital, 3-1 Yoshina, Yamanacho, Sukumo-City 788-0785, Kochi, Japan; michio_kuwahara@hatakenmin.jp (M.K.); k44070099@kochi-u.ac.jp (T.S.); lvpippi0724@yahoo.co.jp (Y.N.); jm-s.shimizu@kochi-u.ac.jp (S.S.); toyokazu_akimori@hatakenmin.jp (T.A.)
2   Department of Pathology, Hata Kenmin Hospital, 3-1 Yoshina, Yamanacho, Sukumo-City 788-0785, Kochi, Japan; mhiroi@mac.com
*   Correspondence: jm-nobtanioka@kochi-u.ac.jp; Tel.: +81-090-1570-9682

**Abstract:** While the importance of conversion surgery has increased with the development of systemic chemotherapy for gastric cancer (GC), reports of conversion surgery for patients with GC with distant metastasis and tumor thrombus are extremely scarce, and a definitive surgical strategy has yet to be established. Herein, we report a 67-year-old man with left abdominal pain referred to our hospital following a diagnosis of unresectable GC. Esophagogastroduodenoscopy and contrast-enhanced abdominal computed tomography (CT) revealed advanced GC with splenic metastasis. A splenic vein tumor thrombus (SVTT) and a continuous thrombus to the main trunk of the portal vein were detected. The patient was treated with anticoagulation therapy and systemic chemotherapy comprising S-1 and oxaliplatin. One year following chemotherapy initiation, a CT scan revealed progressive disease (PD); therefore, the chemotherapy regimen was switched to ramucirumab with paclitaxel. After 10 courses of chemotherapy resulting in primary tumor and SVTT shrinkage, the patient underwent laparoscopic total gastrectomy (LTG) and distal pancreaticosplenectomy (DPS). He was discharged without complications and remained alive 6 months postoperatively without recurrence. In summary, the wait-and-see approach was effective in a patient with GC with splenic metastasis and SVTT, ultimately leading to an R0 resection performed via LTG and DPS.

**Keywords:** gastric cancer; splenic metastasis; splenic vein tumor thrombus; conversion surgery; laparoscopic total gastrectomy; distal pancreaticosplenectomy

## 1. Introduction

Gastric cancer is the sixth most common cancer and the third leading cause of cancer-related deaths worldwide [1]. The prognosis for gastric cancer with distant metastasis is poor [2], and chemotherapy with anticancer or molecular-targeted drugs is considered the mainstay of treatment [3,4]. However, recent developments in chemotherapy have had a remarkable impact on treatment strategies for unresectable metastatic gastric cancer [5].

Conversion surgery is defined as surgical treatment for radical resection after chemotherapy for cancers that were originally unresectable due to technical and/or oncological reasons with distant metastases [6]. Conversion surgery for gastric cancer is a new clinical approach, and several investigators have reported that there is a group of patients who potentially benefit from conversion surgery for gastric cancer with limited conditions such as para-aortic lymphadenopathy, a small number of liver and lung metastases, localized peritoneal dissemination, and solitary organ metastases [7–14]. However, there are few reports of conversion surgery for gastric cancer with distant metastasis and tumor thrombus, and its surgical strategy has not yet been established.

Here, we describe a case of advanced gastric cancer with splenic metastasis and splenic vein tumor thrombosis (SVTT) treated with chemotherapy and successfully performed laparoscopic total gastrectomy (LTG) and distal pancreaticosplenectomy (DPS).

## 2. Case Presentation

A 67-year-old man with left abdominal pain was referred from his local hospital for further evaluation following a diagnosis of gastric cancer with splenic metastasis. Blood tests showed anemia with hemoglobin 11.0 g/dL (normal range: 13.5–17.6 g/dL) but no elevation of tumor markers. Esophagogastroduodenoscopy revealed an elevated tumor with irregular ulceration extending from the middle of the gastric body to the cardia (Figure 1A,B). Pathological examination of the biopsied specimens of the lesion demonstrated medium-differentiated adenocarcinoma, and immunohistochemistry of the tumor showed negative reactivity for human epidermal growth factor receptor 2 (HER2). Contrast-enhanced abdominal computed tomography (CT) revealed irregular wall thickening with a heterogeneous contrast effect in the upper gastric body (Figure 1C). Lymph node swelling around the splenic hilum and numerous irregular masses inside the spleen were observed (Figure 1D).

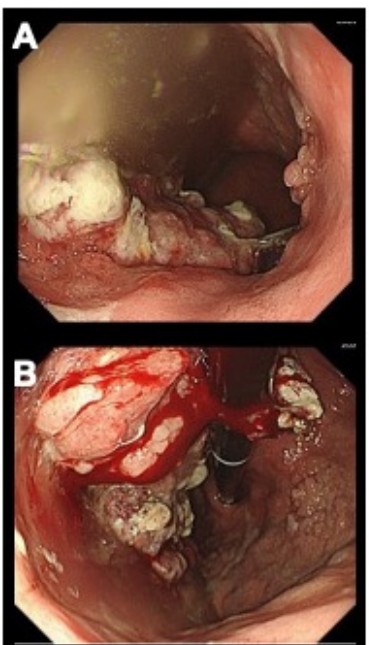 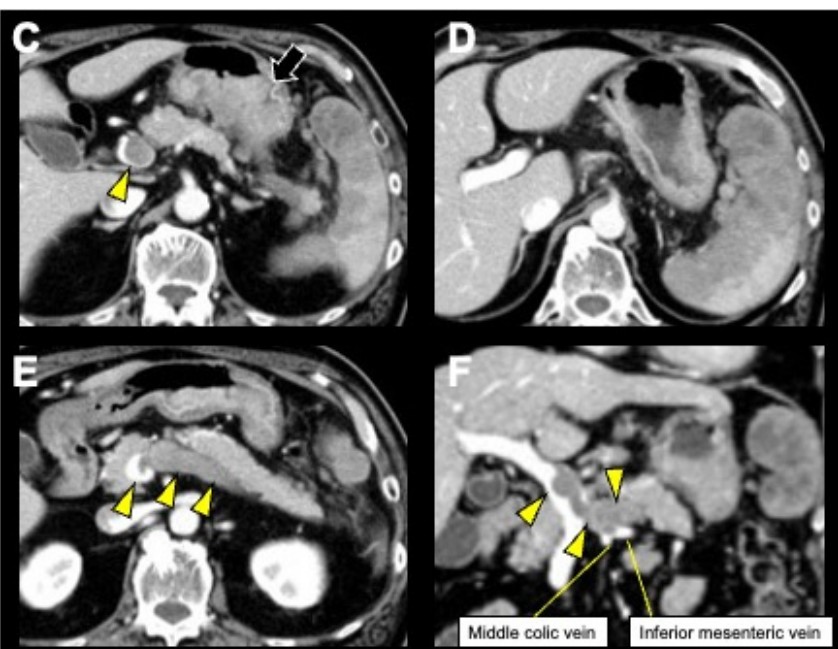

**Figure 1.** Esophagogastroduodenoscopy revealed an elevated tumor with irregular ulceration extending from the middle of the gastric body to the cardia (**A**,**B**). Contrast-enhanced abdominal computed tomography (CT) revealed irregular wall thickening with a heterogeneous contrast effect in the upper gastric body (arrow) (**C**). Lymph node swelling around the splenic hilum and numerous irregular masses inside the spleen were observed (**D**). A tumor thrombus occupying the entire splenic vein and a continuous thrombus extending from the splenic vein confluence to the main trunk of the portal vein were detected (arrowhead) (**C**,**E**,**F**).

Additionally, a tumor thrombus occupying the entire splenic vein and a continuous thrombus from the splenic vein confluence to the main trunk of the portal vein were detected (Figure 1C,E,F). The portal vein thrombus was considered secondary. We diagnosed cT4bN2M1, stage IV gastric cancer (Japanese classification of gastric carcinoma/JCGC 15th) [15] with splenic metastasis, SVTT, and portal vein thrombus.

The patient was treated with anticoagulation with warfarin and systemic chemotherapy with S-1 and oxaliplatin. The patient received oral S-1 (80 mg/m$^2$ twice daily) on days 1–14 and intravenous oxaliplatin (100 mg/m$^2$) on day 1, every 3 weeks. After five

courses, oxaliplatin was discontinued due to intolerable peripheral neuropathy, and S-1 was continued alone. The portal vein thrombus resolved 4 months after the start of warfarin; therefore, anticoagulation therapy was discontinued. One year after the first chemotherapy, esophagogastroduodenoscopy and CT revealed re-enlargement of the primary tumor (Figure 2A–F), which was evaluated to a progressive disease (PD) according to the Response Evaluation Criteria in Solid Tumors (RECIST), version 1 [16]. Therefore, the regimen was switched to ramucirumab (8 mg/kg IVDI on days 1 and 15) in combination with paclitaxel (80 mg/m² IVDI on days 1, 8, and 15) as the second-line treatment.

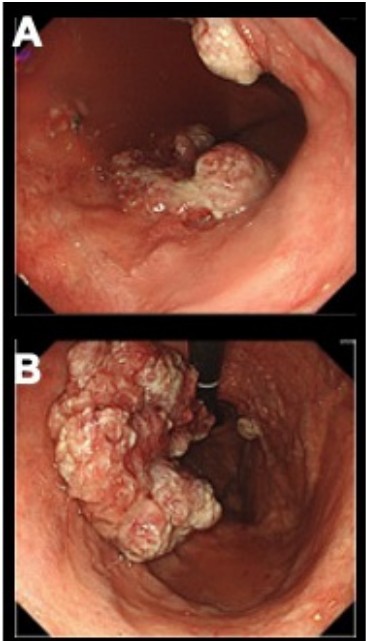 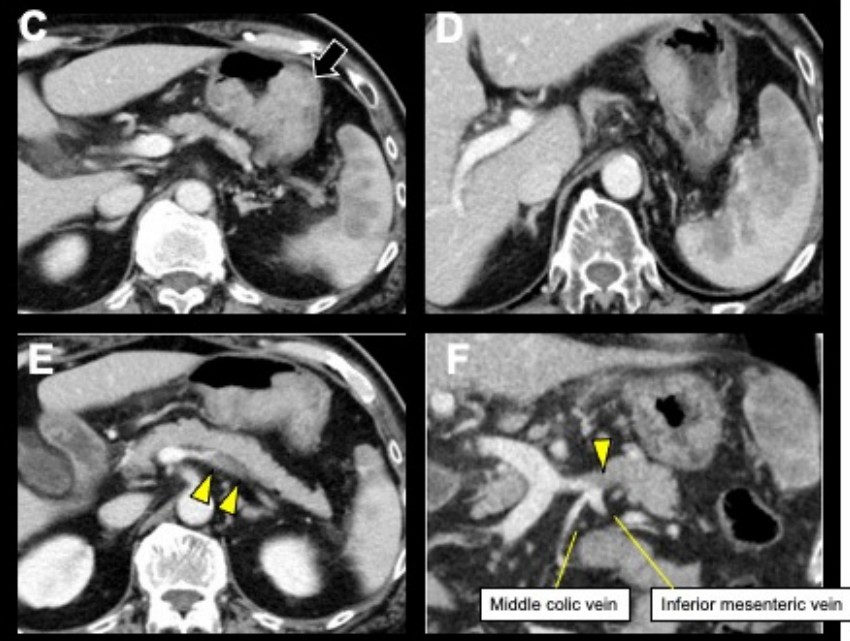

**Figure 2.** Esophagogastroduodenoscopy and contrast-enhanced abdominal computed tomography (CT) showed re-enlargement of the primary tumor (arrow) (**A**–**C**). Splenic metastasis continued to shrink (**D**). The thrombus in the main trunk of the portal vein disappeared and the tumor thrombus in the splenic vein shrunk (arrowhead) (**E**,**F**).

After 10 courses of chemotherapy, the tumor shrank (Figure 3A–C), but it was considered difficult to continue this regimen due to hematologic toxicity. Preoperative contrast-enhanced abdominal CT revealed the absence of a portal vein thrombus (Figure 3D). Although there was a residual tumor in the peripheral splenic vein, the central side from the confluence of the inferior mesenteric and splenic veins was intact (with a secure margin of 12 mm from the portal vein) (Figure 3D,E). After obtaining informed consent from the patient, LTG and DPS were planned for conversion surgery.

Under general anesthesia, the patient was placed in a head-up position. The first port (8 mm) for the camera was inserted at the umbilicus. Four additional ports were inserted: a 5 mm port in the right and left hypothalamus and left side of the abdomen and a 12 mm port in the right side. On exploration, no macroscopic peritoneal dissemination or invasion of gastric cancer into the surrounding organs was observed. Vascular resection of the right side of the stomach with D2 lymph node dissection and duodenal transection were performed. After resecting the left gastric and splenic arteries, the splenic vein was ligated and transected at its confluence with the portal vein. The middle colonic and inferior mesenteric veins that joined the splenic vein were sacrificed. The pancreas was transected at the left margin of the portal vein using an Endo GIA Reinforced Reload with Tri-Staple technology (Medtronic, Dublin, Ireland). The residual lymph nodes and esophagus were resected, and LTG with Roux-en-Y reconstruction and DPS was successfully achieved. In addition, a prophylactic cholecystectomy was performed.

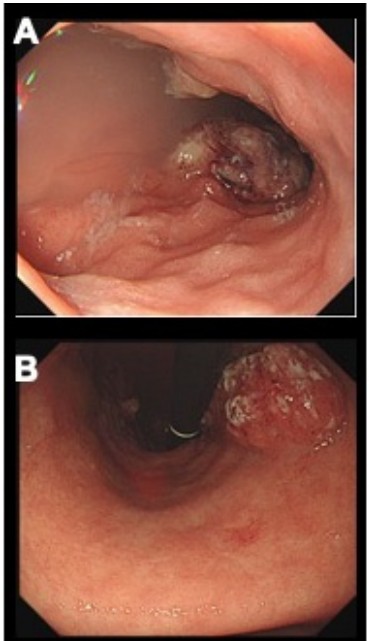
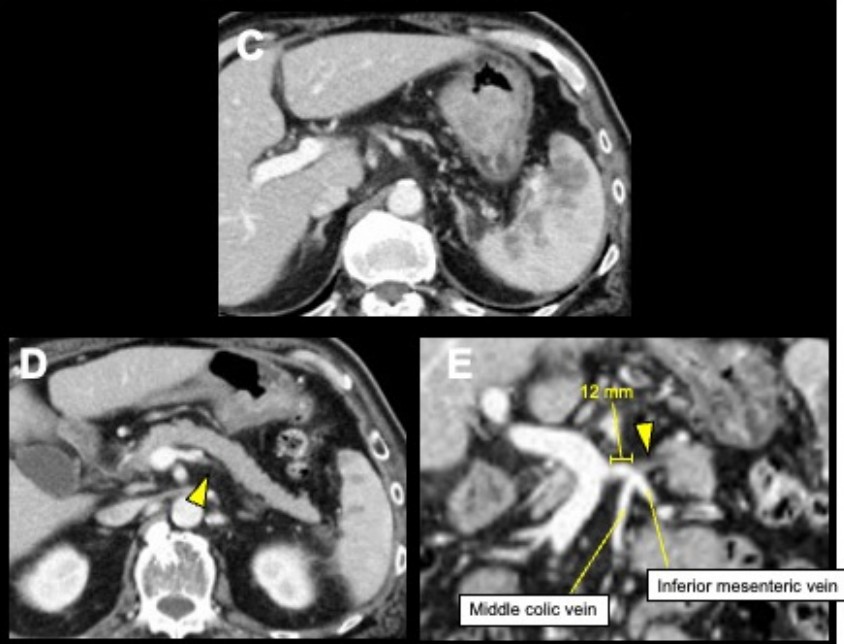

**Figure 3.** Esophagogastroduodenoscopy and contrast-enhanced abdominal computed tomography (CT) showed a reduced primary tumor and splenic metastases (**A–C**). Although there was a residual tumor in the peripheral splenic vein (arrowhead), the central side from the confluence of the inferior mesenteric and splenic veins was intact (with a secure margin of 12 mm from the portal vein) (**D,E**).

Macroscopic findings of the surgical resected specimen revealed a 7.5 cm × 6.0 cm irregularly modified ulcerative lesion in the middle gastric body (Figure 4A,B). Multiple white masses with indistinct borders were observed in the spleen (Figure 4C). Pathology revealed a mixed differentiated adenocarcinoma infiltrating the sub-serosal layer with two lymph node metastases and lymphatic and venous invasion. The splenic vein was occluded up to 5 mm from the cut edge, and fibrosis was observed with adenocarcinoma (Figure 4D,E). No cancer cells were observed near the transected ends of the splenic vein. Cancer cell extension was observed in a vein around the splenic hilum, and an intravenous tumor thrombus was also observed in the short gastric veins. The tumor after preoperative therapy was classified as ypT4N1M1 stage IVB, and the histological response after preoperative therapy was grade 1a, according to the Japanese classification system [15].

The patient was discharged from the hospital within one-week post-surgery without any postoperative complications. The patient did not undergo chemotherapy, and a CT scan performed 6 months after surgery indicated no apparent recurrence.

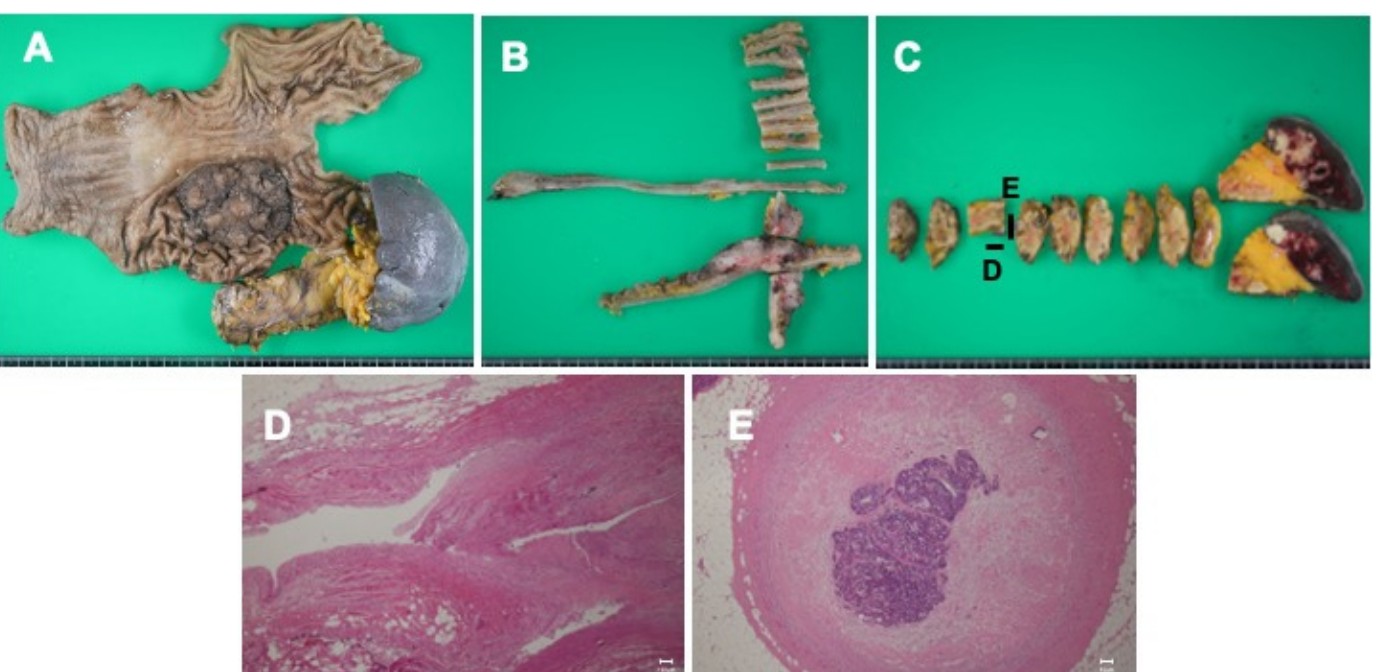

**Figure 4.** Macroscopic findings of the surgical resected specimen revealed a 7.5 cm × 6.0 cm irregularly modified ulcerative lesion in the middle gastric body (**A**,**B**). Multiple white masses with indistinct borders were observed in the spleen (**C**). The splenic vein was occluded up to 5 mm from the cut edge (magnification 40×) (**D**), and fibrosis was observed with adenocarcinoma (magnification 100×) (**E**). No cancer cells were observed near the transected ends of the splenic vein.

## 3. Discussion

In our patient with gastric cancer, two important issues were highlighted: splenic metastasis and SVTT. Because splenic metastasis of gastric cancer is usually associated with peritoneal dissemination or multiple visceral metastases, there are few reports of resection for solitary splenic metastasis [17]. Despite the anatomical proximity between the stomach and spleen, splenic metastasis of gastric cancer is rare. A study of 93 cases of solitary splenic metastasis reported that in 7.5% of cases, gastric cancer metastasis occurred [18]. Gastric cancer causes splenic metastasis through three pathways: (1) via the lymphatic route, (2) via the splenic vein, and (3) via the splenic artery [17]. The spleen has a poorly developed lymphatic system and few afferent lymphatic vessels. Therefore, metastasis via the lymphatic route is rare. The splenic vein route occurs in limited conditions, such as the presence of portal hypertension or liver disease with splenic vein thrombus, because the tumor cells must flow retrogradely through the splenic vein. In the splenic artery route of metastasis, tumor cells enter the spleen via systemic circulation. Thus, splenic metastasis usually occurs as multivisceral organ metastasis [19]. The prognosis of solitary splenic metastasis after curative resection is unclear due to the paucity of reports and short observation period [20]. There have been reports of metastases appearing in other organs after surgery because solitary splenic metastasis was an initial finding of systemic distant metastasis [19].

Tumor thrombus of the portal system is another poor prognostic factor associated with advanced gastric cancer, and chemotherapy is generally the mainstay of treatment [21]. Liver metastases with gastric cancer result from vascular seeding through the gastric drainage vein, which may form a tumor thrombus upstream in the portal or gastric and splenic veins. Eom et al. [22] reported a poor prognosis of 5.4 months for 51 patients with gastric cancer with portal vein tumor thrombus with or without liver metastases.

However, it is also true that in some of these cases, long-term survival can be achieved with multimodality treatment involving radical resection [17,20,21,23]. To our knowledge,

there is only one prior reported case of R0 surgery performed in a patient with splenic metastasis and a tumor thrombus of the portal system [24]. Isigami et al. [24] reported a case of 6-month recurrence-free survival after open total gastrectomy and splenectomy for advanced gastric cancer with splenic metastasis and SVTT after 10 courses of chemotherapy (S-1 and CDDP).

As mentioned above, splenic metastasis and tumor thrombus of the portal system are high-risk conditions for further hematogenous metastasis, and a wait-and-see approach is recommended [20]. In other words, the strategy is to perform surgery in anticipation of R0 resection if the tumor can be controlled after a certain period of chemotherapy without the appearance of systemic metastases. Although there are no clear criteria regarding this waiting period, we decided to proceed with surgery because of the absence of distant metastasis in the long term by systemic chemotherapy. Ishigami et al. reported that only primary gastric cancer and splenic metastasis were resected because the tumor thrombus completely disappeared after preoperative chemotherapy [24]. Nevertheless, considering the presence of residual viable cells in the fibrotic thrombus in our case, it is advisable to perform thorough resection with sufficient margins to improve the success rate of achieving R0 surgery.

There are two problems associated with conversion surgery. First, the extent of resection is larger and it is technically more difficult to achieve radical resection than in conventional surgery. Second, preoperative chemotherapy is often accompanied by a decline in a patient's physical status. Therefore, conversion surgery can be burdensome for patients, with a reported postoperative complication rate of up to 24% [7]. Surgery should be performed safely and minimally invasively, with careful planning based on preoperative imaging.

Minimally invasive surgery for gastric cancer has been established in recent years, and laparoscopic distal gastrectomy for advanced gastric cancer, including after preoperative chemotherapy, was shown to be non-inferior to open surgery in terms of perioperative complications, mortality, and long-term survival [25,26]. Although there are few evidence-based reports on the safety of LTG and LTG+$\alpha$, reports on their efficacy have increased in recent years [27,28]. In the present case, LTG with DPS was successfully performed without postoperative complications. However, given the short observation period, careful follow-up is required to monitor recurrence.

## 4. Conclusions

We performed minimally invasive radical resection for a patient with gastric cancer with splenic metastasis and SVTT through LTG and DPS. As conversion surgery is expected to become increasingly important in the future, a careful preoperative strategy must be developed to achieve radicality and minimal invasiveness.

**Author Contributions:** Conceptualization, N.T.; methodology, N.T.; data acquisition, T.S., Y.N., S.S. and N.T.; writing—original draft preparation, N.T.; writing—review and editing, T.S., Y.N., S.S., M.H., M.K. and T.A.; pathological interpretation, M.H.; supervision, M.K. and T.A; project administration, T.A. All authors have read and agreed to the published version of the manuscript.

**Funding:** This research received no external funding.

**Institutional Review Board Statement:** This study has been performed according to the Declaration of Helsinki. As this paper is a case report, it was exempted from the requirement for ethical approval from the Hata Kenmin Hospital.

**Informed Consent Statement:** Written informed consent was obtained from the patient for publication of this case report and accompanying images.

**Data Availability Statement:** Data sharing is not applicable to this article as no datasets were generated or analyzed during the current study.

**Conflicts of Interest:** The authors declare no conflicts of interest.

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
