# Peer review of "Minimally Invasive Conversion Surgery for Unresectable Gastric Cancer with Splenic Metastasis and Splenic Vein Tumor Thrombus: A Case Report"

_curroncol, doi:10.3390/curroncol31050201_

Round 1

Reviewer 1 Report

Comments and Suggestions for Authors

Thank you the Authors for the opportunity to review this interesting article. Conversion surgery consists of surgical resection with R0 purposes following chemotherapy treatment of lesions initially classified as unresectable. Taking into account the classification of Yoshida et al., unresectable stage IV includes para-aortic lymph node involvement and peritoneal metastases. The clinical case you propose is even rarer as the patient has stage IV gastric cancer due to a solitary splenic metastasis (already rare in itself and not widely described in the literature), SVTT and a continuous thrombus to the main trunk of the portal vein. The result obtained through a minimally invasive approach of laparoscopic total gastrectomy and distal pancreaticosplenectomy is noteworthy.

Obviously the follow-up was short and the cases described are few in the literature. Longer-term results should be evaluated so as to make this minimally invasive approach for stage IV GC a future treatment option.

Author Response

Reviewer 1: Thank you the Authors for the opportunity to review this interesting article. Conversion surgery consists of surgical resection with R0 purposes following chemotherapy treatment of lesions initially classified as unresectable. Taking into account the classification of Yoshida et al., unresectable stage IV includes para-aortic lymph node involvement and peritoneal metastases. The clinical case you propose is even rarer as the patient has stage IV gastric cancer due to a solitary splenic metastasis (already rare in itself and not widely described in the literature), SVTT and a continuous thrombus to the main trunk of the portal vein. The result obtained through a minimally invasive approach of laparoscopic total gastrectomy and distal pancreaticosplenectomy is noteworthy.

Obviously the follow-up was short and the cases described are few in the literature. Longer-term results should be evaluated so as to make this minimally invasive approach for stage IV GC a future treatment option.

 Reply: Thank you for your high recognition of this case report. As you pointed out, the short follow-up period is the limitation in this case, as mentioned at the end of the Discussion section. We believe that the fact that curative resection was achieved for unresectable gastric cancer with a splenic metastasis and SVTT is significant. It is expected that a paper referring to the long-term results will be published in the future. We appreciate your kind understanding.

Reviewer 2 Report

Comments and Suggestions for Authors

I really enjoyed this case report. I think one thing to note is that this also speaks to cancer biology and how we should be operating on more people with metastatic disease who have a good response to therapy (especially if we have markers like ctDNA and other tumor markers to back this up). 

I've never heard the term "conversion surgery". Would say something like "Minimally invasive surgery for initially unresectable gastric cancer..." 

Comments on the Quality of English Language

Several terms are confusing such as "conversion surgery" and "vasectomy". Just need a read through to check wording. 

Author Response

reviewer 2

1. I really enjoyed this case report. I think one thing to note is that this also speaks to cancer biology and how we should be operating on more people with metastatic disease who have a good response to therapy (especially if we have markers like ctDNA and other tumor markers to back this up).

Reply: Thank you very much for your high evaluation of this clinical paper. As you mention, aggressive resection may be recommended even for cancers with distant metastases that are clinically very responsive to chemotherapy. We also believe that the clinical characteristics of the tumors need to be further investigated using liquid biomarkers and other methods.

2. I've never heard the term "conversion surgery". Would say something like "Minimally invasive surgery for initially unresectable gastric cancer..." 

Reply: Thank you for your insightful comments. As you pointed out, I recognized that the term "conversion surgery" needed a detailed explanation at the beginning of this paper.The definition of conversion surgery is discussed by Yoshida et al [PMID: 26643880], and the following has been added to the Introduction section and slightly modified to be more contextual.

…”Conversion surgery is defined as surgical treatment for radical resection after chemotherapy for cancers that were originally unresectable due to technical and/or on-cological reasons with distant metastases [6]. ”…

Please kindly check the revised manuscript.

3. Several terms are confusing such as "conversion surgery" and "vasectomy". Just need a read through to check wording. 

Reply: Thank you for your expert guidance. In particular, there was some confusion of terminology in intraoperative findings, which was corrected after careful consideration. We hope you will kindly check this change in revised manuscript.

Reviewer 3 Report

Comments and Suggestions for Authors

This case report presents conversion surgery for unresectable gastric consisting in  laparoscopic total gastrectomy and distal pancreaticosplenectomy. Considering the rarity of it (distant metastasis and tumor thrombus) and the quality of the presentation I consider it useful for illustrating the new tendencies.

The figures , esophagogastroduodenoscopies and CT performed both initially and after chemotherapy and the images with the specimen and microscopy, are mandatory for a good case report.

The discussions are thorough and present todays level of knowledge.

The 28 references are important and some recent, with no self-citation.

Author Response

Reviewer 3

This case report presents conversion surgery for unresectable gastric consisting in  laparoscopic total gastrectomy and distal pancreaticosplenectomy. Considering the rarity of it (distant metastasis and tumor thrombus) and the quality of the presentation I consider it useful for illustrating the new tendencies.

The figures , esophagogastroduodenoscopies and CT performed both initially and after chemotherapy and the images with the specimen and microscopy, are mandatory for a good case report.

The discussions are thorough and present todays level of knowledge.

The 28 references are important and some recent, with no self-citation.

Reply: Thank you for your encouraging evaluation for our paper and insightful comments. Esophagogastroduodenoscopy  and CT images after initial chemotherapy were added as Figure.2. Please kindly check the revised manuscript.